# Background-Quenched Aggregation-Induced Emission through Electrostatic Interactions for the Detection of Poly(ADP-ribose) Polymerase-1 Activity

**DOI:** 10.3390/molecules28124759

**Published:** 2023-06-14

**Authors:** Fengli Gao, Ruimin Zhao, Liping Huang, Xinyao Yi

**Affiliations:** 1Henan Province of Key Laboratory of New Optoelectronic Functional Materials, College of Chemistry and Chemical Engineering, Anyang Normal University, Anyang 455000, China; 2College of Chemistry and Chemical Engineering, Central South University, Changsha 410083, China

**Keywords:** poly(ADP-ribose) polymerase-1, aggregation-induced emission, background-quenching, electrostatic interactions, fluorescence assay

## Abstract

Poly(ADP-ribose) polymerase-1 (PARP1) is a potential biomarker and therapeutic target for cancers that can catalyze the poly-ADP-ribosylation of nicotinamide adenine dinucleotide (NAD^+^) onto the acceptor proteins to form long poly(ADP-ribose) (PAR) polymers. Through integration with aggregation-induced emission (AIE), a background-quenched strategy for the detection of PARP1 activity was designed. In the absence of PARP1, the background signal caused by the electrostatic interactions between quencher-labeled PARP1-specitic DNA and tetraphenylethene-substituted pyridinium salt (TPE-Py, a positively charged AIE fluorogen) was low due to the fluorescence resonance energy transfer effect. After poly-ADP-ribosylation, the TPE-Py fluorogens were recruited by the negatively charged PAR polymers to form larger aggregates through electrostatic interactions, thus enhancing the emission. The detection limit of this method for PARP1 detection was found to be 0.006 U with a linear range of 0.01~2 U. The strategy was used to evaluate the inhibition efficiency of inhibitors and the activity of PARP1 in breast cancer cells with satisfactory results, thus showing great potential for clinical diagnostic and therapeutic monitoring.

## 1. Introduction

Poly-ADP-ribosylation (PARylation) is an important post-translational modification in cellular physiology. As the leading PARylation enzyme, poly(ADP-ribose) polymerase-1 (PARP1) is involved in many cellular processes, such as DNA repair, transcriptional regulation and signal transduction [1,2]. The activity of PARP1 is closely related to tumorigenesis and malignant progression and its inhibitors have a wide spectrum of applications in diagnostics and therapeutics [3]. It has been documented that PARP1 can be regarded as a potential biomarker and therapeutic target for ischemic diseases, cardiac hypertrophy, diabetes, inflammation or neuronal death and some cancers (e.g., ovarian, breast and oral) [4,5]. Therefore, it is of great importance to develop simple, sensitive and accurate methods for probing PARP1 activity and screening its inhibitor drugs.

After binding to damaged DNA, quadruplexes or specific double-stranded DNA (dsDNA), PARP1 can be activated to catalyze the PARylation of nicotinamide adenine dinucleotide (NAD^+^) onto the acceptor proteins, including histone, transcription factor and PARP1 itself [6,7]. The resulting poly(ADP-ribose) (PAR) polymer contains 50–200 ADP-ribose units in a linear or branched format. Although the development of PAR antibodies and NAD^+^ analogs such as radio-labeled, biotinylated, isotope-labeled and clickable NAD^+^ derivatives has promoted the emergence of various technologies for PARP1 detection [8,9,10], most of the methods require radioactive labeling and expensive regents. To overcome these limitations, various novel heterogeneous and homogeneous strategies have been explored for the detection of PARP1 activity and screening of potential inhibitors, including electrochemistry, photoelectrochemistry, fluorescence, colorimetry and quartz crystal microbalance (QCM) [11,12,13,14,15,16,17,18]. Heterogeneous assays, which involve immobilizing the target-specific DNA on a solid interface to capture and activate PARP1, exhibit the advantages of signal amplification, less sample demand and high sensitivity as well as selectivity. Thus, several groups have used DNA-modified electrodes to capture and activate PARP1 for the follow-up PARylation [19,20,21]. The resulting PAR polymers could be recognized by positively charged [Ru(NH_3_)_6_]^3+^, polyanilne and peptide-templated copper nanoparticles, thus allowing for the signal-amplified electrochemical detection of PARP1. Meanwhile, positively charged gold nanorods, gold nanoclusters and cationic conjugated polymers have been used as the labels for PARP1 detection in QCM, chemiluminescence and photoelectrochemistry, respectively [22,23,24]. These heterogeneous methods show high sensitivity and specificity. However, as they require sophisticated modification procedures, the attachment of PARP1 on the solid surface may limit polymerization efficiency due to steric hindrance, and PARylation will cause the subsequent dissociation of PARP1 from DNA [25,26]. In addition, most of the methods are based on the electrostatic interactions between the positively charged signal probes and the negatively charged ADP-ribose units, which will produce false-positive signals and a low signal-to-noise ratio due to the electrostatic interactions between the signal probes and the DNA strands. On the contrary, homogeneous assays for monitoring enzymatic activity have the merits of easy operation, high throughput and high efficiency [27,28]. As a homogenous method, fluorescent analysis is the most popular due to its inherent advantages of high simplicity, sensitivity and accuracy. The emission properties of fluorophores may vary greatly in different environments. For example, some fluorescence probes and nanoparticles can differentiate double-stranded DNA (dsDNA) from single-stranded DNA (ssDNA) [29,30,31]. Recently, Yang et al. found that TOTO-1 (a cyanine dye dimer) can identify G-base-rich ssDNA and PAR polymer, and developed a homogenous fluorescent method for the detection of PARP1 [31]. However, the method requires the use of Exo III to reduce the background signal by digesting the dsDNA substrates used for the capture and activation of PARP1. Thus, a simple, sensitive and accurate method for probing PARP1 activity and screening potential inhibitors is needed.

Fluorogens with aggregation-induced emission (AIE) properties are non-emissive in the dissolved or dispersed state but will emit strong fluorescence in the form of supramolecular aggregates [32]. Unlike aggregation-caused quenching (ACQ) fluorophores, fluorogens with AIE features have a remarkable optical property and strong resistance toward photobleaching [33]. Such materials have been used in a broad range of fields, including organic optoelectronic devices, biosensing, bioimaging, photodynamic therapy and photothermal therapy [34,35,36,37]. By conjugating fluorogens with the target ligands, many AIE chemo/biosensors have been designed for the determination of various analytes, such as ions, small molecules, microenvironment sensing, biological macromolecules, cellular processes and pathogens [38,39,40]. The aggregation of AIEgens can be triggered by different stimuli, including solubility change, hydrophobic assembly, hydrogen bonding, target–receptor binding and electrostatic interactions. It has been documented that negatively charged DNA polymers can induce the assembly of some positively charged fluorogens to activate the AIE process [41,42,43,44]. For example, tetraphenylethene-substituted pyridinium salt (TPE-Py) is a positively charged AIE molecule with long-wavelength emission and high fluorescence quantum yield that has been used to design AIE-based fluorescence methods by means of electrostatic interactions [45]. Zhuang et al. reported the photostable imaging of activated telomerase in living cells with TPE-Py as the AIE fluorogen [43]. Moreover, the in vitro detection and in vivo probing of enzyme activity has also been achieved using fluorogens to modify DNA or peptide probes [46,47]. For instance, Min et al. monitored telomerase activity with TPE-Py-modified DNA as the fluorescence reporter and Exo III as the signal amplifier. Qi and co-workers achieved the detection of Cu^2+^ in aqueous solution and living cells using a TPE-Py-modified peptide probe. Herein, we suggest that TPE-Py could be used to develop AIE-based biosensors for the detection of PARP1 activity by integrating the fluorescence resonance energy transfer (FRET) effect to reduce the background signal (Figure 1). Negatively charged DNA hybrids could induce the aggregation and emission of positively charged TPE-Py fluorogens through electrostatic interactions. However, the emission was efficiently quenched by FRET from the TPE-Py aggregates to the quenchers labeled at the dsDNA probes (Q_2_-dsDNA). After PARylation, the resulting PAR with a large number of negatively charged ADP-ribose units would attach TPE-Py through electrostatic assembly. The aggregates attached to PAR polymers are far away from the quenchers, thus lighting up the fluorescence. The homogeneous method does not require the use of a labeled NAD^+^ analog as the substrate and the additional procedures to reduce the background signal by digesting the DNA probes, thus exhibiting the advantages of high simplicity, low cost and rapid response.

## 2. Results and Discussion

### 2.1. AIE Propensity of TPE-Py

Owing to its reliable AIE performance and facile synthesis with various derivatizations, TPE has been extensively utilized as the basic scaffold in various applications. In this work, TPE-Py, a simple molecule consisting of a pyridinium group and a typical TPE unit, was employed as the AIEgen. To probe the AIE behavior of TPE-Py, the fluorescence spectra of TPE-Py in the solvent–water mixed solvent were collected. As shown in Figure 1A, TPE-Py in pure DMSO solution emitted a red emission at 625 nm (curve a). The emission gradually weakened with an increase in water content up to 90% (curve b and c). This was due to the intramolecular charge transfer (ICT) effect caused by the interaction between the electron-donating TPE unit and the electron-accepting pyridinium moiety [48]. The self-assembly behavior of TPE-Py in low solvent content was investigated using transmission electron microscope (TEM) and dynamic light scattering (DLS) (Figure 1C,D). Small nanoparticles were clearly observed with TEM, which is indicative of the self-assembly of TPE-Py molecules. The nanoparticles showed weak emission probably due to the ICT effect [48,49,50]. The DLS analysis further indicated that the nanoparticles were dispersed and possessed high positive charges (zeta potential = 5.7), which could be ascribed to the quaternary ammonium salt groups on their surface. The charge became −6.7 after the addition of the negatively charged dsDNA. Meanwhile, the size of TPE-Py aggregates was greatly increased. This result can be attributed to the further assembly of nanoparticles into larger aggregates, which was confirmed through TEM observation and DLS analysis (Figure 1C,D). Moreover, the negatively charged dsDNA lit up the fluorescence of TPE-Py AIEgens at around 550 nm, as shown in Figure 1B (curve a). In this process, the restriction of the intramolecular motion (RIM) process occurred and the radiation-less relaxation pathway of TPE-Py was blocked. Nevertheless, dsDNA-induced emission through electrostatic interactions may cause a relatively high background signal and an unsatisfactory signal-to-background ratio, thus reducing the sensitivity for PARP1 detection. To solve this problem, two DNA strands were individually labeled with a traditional fluorescent quencher (dabcyl (4-(4-(dimethylamino)phenylazo)benzoic acid)) (Figure 1). The fluorescence of the mixture of TPE-Py and DNA formed with dual-labeled dsDNA (Q_2_-dsDNA) (curve c) was significantly lower than that formed with label-free dsDNA (curve a) and single-labeled dsDNA (Q_1_-dsDNA) (curve b). This suggests that the fluorescence of TPE-Py nanoparticles electrostatically adsorbed on the backbones of dsDNA could be quenched via the FRET mechanism due to the overlap between the emission of TPE-Py and the absorption of dabcyl. Thus, labeling dsDNA with two quenchers could decrease the background signal. Since the FRET effect is dependent on the distance between the quenchers and luminophores [42,43,44], TPE-Py AIEgens assembled on the resulting PAR polymers via electrostatic interactions would be far away from the dabcyl groups, thus lighting up the fluorescence.

### 2.2. Feasibility for PARP1 Detection

After the activation of PARP1 by the co-factor dsDNA, PARP1 could act as the catalytic molecule to cleave NAD^+^ into nicotinamide and ADP-ribose unit and then cause the formation of PAR polymers with many negatively charged ADP-ribose units. In the presence of NAD^+^, the fluorescence signal of TPE-Py showed negligible change. Thus, positively charged TPE-Py could potentially be used as the AIEgen to measure the activity of PARP1 by monitoring the formation of PAR polymers with a large number of negative charges. To evaluate the feasibility of the AIE-based method, PARP1 was analyzed with dsDNA and Q_2_-dsDNA as the co-factor, respectively. As shown in Figure 2, incubation of PARP1 with NAD^+^/dsDNA or NAD^+^/Q_2_-dsDNA caused an increase in the fluorescence intensity of TPE-Py to some extent. The increasing multiple in the NAD^+^/Q_2_-dsDNA system was 1351%, which is higher than that in the NAD^+^/dsDNA system (157%). This implies that the background-quenched strategy achieves higher sensitivity. In contrast to the previously reported fluorescence method [31], the homogeneous strategy does not require the use of additional reagents or procedures to reduce the background signal resulting from dsDNA, thus simplifying the detection operation.

### 2.3. Sensitivity for PARP1 Detection

To investigate the sensitivity of our method, different concentrations of PARP1 were determined, and the results are displayed in Figure 3A. The signals were intensified as the concentration of PARP1 increased from 0 U to 20 U. This result implies that a higher concentration of PARP1 favors the conjugation of more ADP-ribose units to PARP1 through intermolecular auto-PARylation. The resulting PAR polymers, with a significant increased charge density, could adsorb many positively charged TPE-Py molecules, eventually leading to an increase in the AIE emission of TPE-Py. As shown in Figure 3B, there was a good linear relationship between the relative change in the fluorescence intensity (defined as (F/F_0_) − 1, where F_0_ and F represent the fluorescence intensity of the system in the absence and presence of PARP-1, respectively) and PARP1 concentration in the range of 0.01~2 U. The linear equation can be described as Y = 0.22 + 4.17X with R^2^ = 0.996. The limit of detection (LOD) was calculated to be 0.006 U (S/N= 3). Compared with methods used in previous works, the sensitivity of this method was comparable or even higher (Table 1). The high sensitivity is attributed to the quenched background and the excellent AIE property of TPE-Py. Although some heterogeneous methods, such as electrochemistry and photoelectrochemistry, show higher sensitivity than our method, the reported heterogeneous assays involve the use of nanomaterials or enzymes for signal amplification and required sophisticated modification procedures. The LOD of this work was comparable to that achieved using a cyanine dye dimer of TOTO-1 to identify the produced PAR polymers in which Exo III was used to digest the dsDNA substrates [31]. Overall, our method exhibited high simplicity and low cost since it did not require the immobilization of dsDNA and PARP1 on the solid surface or the use of nucleases to eliminate the background results.

### 2.4. Evaluation of Inhibition Efficiency

Evaluation of the efficiency of potential inhibitors is of great importance in the exploration of original and effective anti-cancer medicine. To further evaluate the validity of the method, the inhibition efficiency of AG014699 (a classical inhibitor for PARP1) was determined. As displayed in Figure 4, the fluorescence intensity decreased with an increasing amount of AG014699. The derived IC_50_ value for 2 U PARP1 was calculated to be 6.3 nM. This result was in accordance with that of previous works [15,19,22,23]. Thus, the proposed method possesses great potential to screen novel inhibitor drugs for PARP1-related diseases.

### 2.5. Selectivity and Lysate Sample Analysis

To investigate the selectivity, experiments were performed with other biological species as interferences, including HSA, avidin, thrombin and glucose. As shown in Figure 5, the solutions remained nearly non-emissive in the presence of the interferences. Their coexistence did not interfere with the detection of PARP1. These results demonstrate that the proposed method showed excellent selectivity for the detection of PARP1 activity.

To assess its potential for clinical applications, the method was applied to determine the levels of PARP1 in the lysate of breast cancer cells of MCF-7 (Figure 6). The relative change in fluorescence signal increased gradually with an increasing number of MCF-7 cells, indicating that the method could be employed to determine PARP1 levels in lysates. However, there was no significant increase in the signal for the assays of lysates spiked with the inhibitor, further confirming that PARP1 loses its auto-PARylation ability when its activity is inhibited. These results indicate that the method could be utilized to evaluate PARP1 activity in biological samples and screen potential drugs for PARP1-related diseases.

### 2.6. Cell Imaging

PARP-1 is present in normal human cells at very low levels. However, its activity and level are up-regulated in some living cancer cells, especially in the nucleus of cells such as SK-BR-3 and MCF-7. In view of the exciting results of the method for PARP1 detection in lysate samples, its capability for imaging was investigated with living MCF cells as the model cell targets. Prior to intracellular imaging investigation, dsDNA, NAD^+^ and PARP1 were transfected into the cells since no bright fluorescence was observed without the transfection of PARP1. As shown in Figure 7A, yellow-emissive fluorescence images were observed. As a negative control, the PARP1 inhibitor was collectively transfected into the cells with PARP1, and the visual field of fluorescence images remained weak emission (Figure 7B). The results suggest that the method can be used for probing intracellular PARP1 activity, thus providing a promising way of screening inhibition drugs and monitoring therapeutic efficiency.

## 3. Experimental Section

### 3.1. Chemicals and Reagents

NAD^+^, human serum albumin (HSA), avidin, thrombin and AG014699 inhibitor were purchased from Sigma-Aldrich Co., Ltd. (Shanghai, China). PARP1 was ordered from AmyJet Scientific Inc. (Wuhan, China). TPE-Py was ordered from Qiyue Biology Co., Ltd. (Xi’an, China). Ethylene glycol-bis-(aminoethylether)-N,N,N′,N′-tetraacetic acid (EGTA), 3-[(3-cholamidopropyl) dimethylammonio]-1-propanesulfonate (CHAPS), phenylmethylsulfonyl fluoride (PMSF) and tris(hydroxymethyl)aminomethane (Tris) were ordered from Sangon Biotech. Co., Ltd. (Shanghai, China). The label-free DNA strands (5′-CGA GTC TAC AGG GTT GCG GCC GCT TGG G-3′ and CCC AAG CGG CCG CAA CCC TGT AGA CTC G) and the dabcyl-labeled complementary DNA strands (5′-dabcyl-CGA GTC TAC AGG GTT GCG GCC GCT TGG G-3′ and 5′-dabcyl-CCC AAG CGG CCG CAA CCC TGT AGA CTC G-3′) were synthesized and purified by Sangon Biotech. Co., Ltd. (Shanghai, China). Their sequences were designed according to the reference [25]. Other chemicals were provided by Aladdin Reagent Co., Ltd. (Shanghai, China). All solutions were prepared with ultrapure water from a Millipore Milli-Q water system.

The label-free dsDNA duplexes were prepared by incubating two 10 μM complementary DNA strands, 5′-CGA GTC TAC AGG GTT GCG GCC GCT TGG G-3′ and 5′-CCC AAG CGG CCG CAA CCC TGT AGA CTC G-3′, in 10 mM pH 7.4 Tris buffer containing 10 mM NaCl at 37 °C for 60 min. The single-labeled dsDNA (Q_1_-dsDNA) duplexes were prepared by incubating 10 μM 5′-dabcyl-CGA GTC TAC AGG GTT GCG GCC GCT TGG G-3′ and equivalent 5′-CCC AAG CGG CCG CAA CCC TGT AGA CTC G-3′ under the same conditions. The dual-labeled dsDNA (Q_2_-dsDNA) duplexes were prepared through the incubation of 5′-dabcyl-CGA GTC TAC AGG GTT GCG GCC GCT TGG G-3′ and 5′-dabcyl-CCC AAG CGG CCG CAA CCC TGT AGA CTC G-3′.

### 3.2. Instruments

The fluorescence spectra were recorded on a F4600 fluorescence spectrometer (Hitachi, Tokyo, Japan) with an excitation wavelength of 405 nm. The morphology and size of TPE-Py fluorogens were characterized using FEI Tecnai G2 T20 TEM and Nano ZS90 DLS analyzer.

### 3.3. Characterization of TPE-Py

A total of 10 μL of 1 mM TPE-Py in DMSO was diluted to 10 μM using DMSO/Tris mixed solvent with different water fractions. To investigate the DNA-responsive AIE of TPE-Py, 10 μL of 10 μM dsDNA, Q_1_-dsDNA or Q_2_-dsDNA was added to 1 mL of TPE-Py in 10% DMSO buffer solution. After incubation for 10 min, the fluorescence spectra were recorded on the fluorescence spectrometer. The nanoparticles/aggregates were characterized using TEM and DLS analyzer.

### 3.4. Assays of PARP1 Activity

A total of 10 μL of PARP1 at a given concentration was added to 80 μL of reaction buffer (pH 7.4, 10 mM Tris, 10 mM KCl, 1 mM MgCl_2_, 50 μM Zn(OAc)_2_) containing 1 μM Q_2_-dsDNA and 100 μM NAD^+^. After incubation at 37 °C for 60 min, 10 μL of TPE-Py in DMSO was introduced into the reaction solution. After that, the reaction solution was analyzed using the fluorescence spectrometer.

To evaluate the inhibition efficiency of the inhibitor, different concentrations of AG014699 were incubated with PARP1 for 10 min. Other experiments were conducted with procedures similar to those used for the assays of PARP1 in the absence of an inhibitor. The inhibition efficiency (%) was calculated as inhibition (%) = 100 × (F − F′)/(F − F_0_), where F’ is the fluorescence intensity in the presence of 2 U PARP1 with different concentrations of inhibitor.

### 3.5. Assays of PARP1 in Cellular Lysate

The lysate was extracted with the previously reported method [15]. In brief, MCF-7 cells were cultured in Dulbecco’s modified Eagle’s medium (DMEM) supplemented with 10% fetal bovine serum at 37 °C in a humidified atmosphere containing 5% CO_2_. After incubation for 24 h, the cells were washed with phosphate buffer and then lysed with the lysis solution (10 mM Tris, 1 mM MgCl_2_, 1 mM EGTA, 0.1 mM PMSF, 0.5% CHAPS and 10% glycerol). The cell suspension was centrifuged at 12,000 rpm and the supernatant was collected and stored at −80 °C. Before the assays, the lysate was diluted by different folds and then added to the Q_2_-dsDNA- and NAD^+^-containing reaction buffer. Other experiments were conducted with procedures similar to those used for the assays of PARP1 standard samples.

### 3.6. Cell Imaging

MCF-7 cells were incubated in 20 mm confocal dishes for 24 h at 37 °C. Then, 0.5 mL of culture medium containing 1 μM Q_2_-dsDNA, 100 μM NAD^+^ and 10 U PARP1 was added to the cells for 24 h of incubation. In the control group, 50 nM inhibitor (AG01469990) was included in the culture medium. After incubation for 24 h in the cell cultivation environment, the cells were washed twice with the phosphate buffer. This was followed by adding 0.5 mL of 10 μM TPE-Py in 10% DMSO into the confocal dishes. After incubation at 37 °C for 30 min, the cells were washed three times with the phosphate buffer and then imaged on an Olympus IX73 inverted fluorescence microscope. The images were acquired using a 10 × or 20 × oil immersion objective lens.

## 4. Conclusions

In summary, we proposed a homogeneous method for the determination of PARP1 activity based on the background-quenched AIE mechanism using electrostatic interactions. The TEM and DLS results indicate that the negatively charged DNA could trigger the assembly of TPE-Py nanoparticles into large aggregates through electrostatic interactions, resulting in a high background signal. Interestingly, the background could be effectively eliminated through the use of a quencher-labeled dsDNA probe based on the FRET effect. This method showed high sensitivity and could be used to determine the activity of PARP1. It was also used to evaluate the inhibition efficiency and conduct cell imaging with satisfactory results. In contrast to the homogeneous fluorescence method, which uses Exo III to reduce the background signal by digesting dsDNA substrates, our proposal is relatively simple, rapid and cost-effective for assays of PARP1 activity. Although the sensitivity of this method was slightly lower than that of some heterogeneous methods, our proposal showed high simplicity, low cost and rapid response since it did not require the immobilization of dsDNA and PARP1 on the solid surface. Based on the unique features of AIEgens toward negatively charged phosphate bones, the AIE strategy could be employed for the design of various novel biosensors by monitoring the formation or degradation of nucleic polymers.

## Data Availability

Not applicable.

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
