# Peer review of "Background-Quenched Aggregation-Induced Emission through Electrostatic Interactions for the Detection of Poly(ADP-ribose) Polymerase-1 Activity"

_molecules, 2023, doi:10.3390/molecules28124759_

Round 1

Reviewer 1 Report

Overall, this manuscript provides a homogeneous method for the detection of PARP1 activity by AIE. The use of quencher-labeled dsDNA greatly reduced the background, thus improving the sensitivity of this method. This work can be considered to be published in Molecules. However, the following concerns should be addressed before publication.

1. In the first paragraph of introduction, the authors write “It has been documented that PARP1 can be regarded as a potential biomarker and therapeutic target”. A potential biomarker and target for what disease? This is not clear.

2. In the introduction part, the authors write “procedures and expensive detection costs. To overcome these limitations, various label free heterogeneous and homogeneous strategies have been explored for the detection…” However, the following sentences, the authors introduced gold nanorods/nanoparticles as labels in PARP1 detection, which is confusing. Please rewrite this paragraph.

3. The data presentation in Figure 1(A) is confusing. Curves a, b, and c are the free TPE-Py test in different buffer conditions; while d, e, f are after dsDNA addition and tested in the same buffer. Technically, these are two groups of experiments, and it is better to plot them in two panels (with the same scale). Figure 1(C), please use identical names as Y-axis labels. In addition, why was NAD+ added in this assay? Please explain.

4. How did the authors design the dsDNA? Will the sequence and length affect the detection in this study?

5. The cell imaging experiment is too simple. The story is about background-quenched AIE for PARP1 detection. However, the data did not show the comparison between the samples with and without quenchers. Also, there is no comparison between normal cells and cancer cells.

6. All the abbreviations in the manuscript should be carefully checked. For example, in the introduction section, poly(ADP-ribose) was abbreviated (as PAR) in the second paragraph, however, the full name still showed up in the following sentences/paragraphs. Besides, please use QCM for quartz crystal microbalance. Other abbreviations like PARylation, dsDNA, TEM, DLS…

7. Extensive editing of English language in the manuscript is required.

Extensive editing of English language in the manuscript is required.

Author Response

We thank the reviewer for his/her positive comments: Overall, this manuscript provides a homogeneous method for the detection of PARP1 activity by AIE. The use of quencher-labeled dsDNA greatly reduced the background, thus improving the sensitivity of this method. This work can be considered to be published in Molecules. However, the following concerns should be addressed before publication.

Comment 1: In the first paragraph of introduction, the authors write “It has been documented that PARP1 can be regarded as a potential biomarker and therapeutic target”. A potential biomarker and target for what disease? This is not clear.

Response: PARP1 can be regarded as a potential biomarker and therapeutic target for ischemic diseases, cardiac hypertrophy, diabetes, inflammation or neuronal death, and some cancers (e.g. ovarian, breast and oral). We have revised the presentation and cited the references in the first paragraph of introduction.

Comment 2: In the introduction part, the authors write “procedures and expensive detection costs. To overcome these limitations, various label free heterogeneous and homogeneous strategies have been explored for the detection…” However, the following sentences, the authors introduced gold nanorods/nanoparticles as labels in PARP1 detection, which is confusing. Please rewrite this paragraph.

Response: It is a good comment. We have revised the presentation carefully.

Comment 3: The data presentation in Figure 1(A) is confusing. Curves a, b, and c are the free TPE-Py test in different buffer conditions; while d, e, f are after dsDNA addition and tested in the same buffer. Technically, these are two groups of experiments, and it is better to plot them in two panels (with the same scale). Figure 1(C), please use identical names as Y-axis labels. In addition, why was NAD+ added in this assay? Please explain.

Response: We have revised the figures and labels. NAD+ is the substrate of PARylation; it was not included in the experimental of Figure 1. We have revised the figure caption.

Comment 4: How did the authors design the dsDNA? Will the sequence and length affect the detection in this study?

Response: The sequences of dsDNA were designed according to the reference (Nat. Commun., 2020, 11, 2174). Their interaction with PARP1 has been investigated in the report.

Comment 5: The cell imaging experiment is too simple. The story is about background-quenched AIE for PARP1 detection. However, the data did not show the comparison between the samples with and without quenchers. Also, there is no comparison between normal cells and cancer cells.

Response: We thank the reviewer for his/her comments. PARP-1 is present in normal human cells at very low levels. Its activity and level may be up-regulated in some of living cancer cells, especially in the nucleus of cells such as SK-BR-3 and MCF-7. In view of the exciting result of the method for PARP1 detection in lysate samples, the capability for imaging was investigated with living MCF cells cultured in our lab as the model cell targets. Prior to intracellular imaging investigation, PARP1 was transfected into the cells since no bright fluorescence was observed without the transfection of PARP1. We have discussed the results in the revised manuscript.

Comment 6: All the abbreviations in the manuscript should be carefully checked. For example, in the introduction section, poly(ADP-ribose) was abbreviated (as PAR) in the second paragraph, however, the full name still showed up in the following sentences/paragraphs. Besides, please use QCM for quartz crystal microbalance. Other abbreviations like PARylation, dsDNA, TEM, DLS…

Response: We have checked and revised the abbreviations carefully.

Comment 7: Extensive editing of English language in the manuscript is required.

Response: We have asked an English speaker to revise manuscript.

Reviewer 2 Report

The authors submitted a paper named “Background quenched aggregation induced emission by elec-trostatic interactions for the detection of poly (ADP ribose) polymerase 1 activity” which needs further improvements:

The abstract needs to show the numerical results for the study.

Please indicate the novelty.

Keywords need to be revised.

Last paragraph of the study need to show the novel sides and main aim.

Please avoid multiple citations especially in introduction.

What is difference between scheme and figure?

What is experimental section before conclusions?

All figures in results need to be commented deeply.

Please add nomenclature.

english is good

Author Response

We thank the reviewer for his/her positive comments: The authors submitted a paper named “Background quenched aggregation induced emission by electrostatic interactions for the detection of poly (ADP ribose) polymerase 1 activity” which needs further improvements:

Comment 1: The abstract needs to show the numerical results for the study.

Response: We have added the detection limit and linear range of this method in the abstract.

Comment 2: Please indicate the novelty.

Response: We have highlighted the novelty of this work in introduction and conclusion.

Comment 3: Keywords need to be revised.

Response: We have revised the keywords.

Comment 4: Last paragraph of the study need to show the novel sides and main aim.

Response: We have revised last paragraph to highlight the novelty of this work.

Comment 5: Please avoid multiple citations especially in introduction.

Response: We have updated the references.

Comment 6: What is difference between scheme and figure?

Response: Scheme generally refers to the principle of the proposal, while figure mostly refers to the spectra and picture for the experimental results. We have checked the figure numbers and captions carefully.

Comment 7: What is experimental section before conclusions?

Response: According to regulations for Molecules journal, “Results” should be before “Materials and Methods”.

Comment 8: All figures in results need to be commented deeply.

Response: We have discussed the results deeply and added some presentations in the revised manuscript.

Comment 9: Please add nomenclature.

Response: We have checked and revised the abbreviations carefully. The abbreviations were used according to the references.

Comment 10: english is good.

Response: We thank the reviewer for his/her positive comments.

Round 2

Reviewer 1 Report

The revised the manuscript has already addressed the concerns in the previous review comments. The description on the data matches the experimental design. In addition, more details have been added in the results section, which provides more comprehensive discussion. Overall, the revised manuscript is suitable for publication in Molecules.